# Cell-painting phenotyping in filamentous fungi: deep learning for image-based drug discovery

**Polina Simkina** [1]                                    POLINA.SIMKINA@BAYER.COM
**Bruno Leggio** [1]                                       BRUNO.LEGGIO@BAYER.COM
**Gerard Sanroma-Guell** [2]                     GERARD.SANROMA-GUELL@BAYER.COM
**Yoann Huet** [1]                                           YOANN.HUET@BAYER.COM

[1] *R&D Disease Control, Bayer SAS, Crop Science Division, Lyon, France*

[2] *Engineering & Technology, Computer Vision Innovation, Bayer AG, Leverkusen, Germany*

**Editors:** Under Review for MIDL 2024

## Abstract

The advancement of Cell Painting, a relatively new morphological profiling assay, requires adequate and effective image analysis methods suitable for high-content data. That is why, developing deep learning methods for biological image analysis is extremely relevant. In this paper, we present feature extraction results on Cell Painting images of phytopathogenic fungi using deep learning techniques. We give an overview of the current method, which is based on a supervised convolutional neural network, and present the prospects and advantages of leveraging a self-supervised model based on vision transformers in the given problem setting.

**Keywords:** Drug discovery, Biological Image Analysis, Self-supervised models.

## 1. Introduction

In morphological profiling for drug discovery, quantitative features are extracted to identify phenotypical characteristics of samples treated with various compounds. Cell Painting (Bray et al., 2016) is a relatively new morphological profiling assay, where up to 6 fluorescent dyes are used to stain major cell compartments and subsequently retrieve relevant biological information using high-throughput microscopy. The advantage of this method is that it offers accurate insights into sub-cellular compartments, bringing detailed information about the effects of screened drugs.

This technique produces large amounts of high-content data, whose relevant information is often hidden under layers of complex and subtle correlated effects. Thus, it is of the utmost importance to develop sophisticated image analysis techniques that allow the extraction of meaningful insights associated with biological processes of interest. In this context, we focus on developing deep learning models for the analysis of biological images, which benefits the precision of morphology-based drug profiling. Several well-established computer vision models have been already explored for Cell Painting in mammalian cells (e.g. (Moshkov et al., 2024), (Kim et al., 2023)), demonstrating a prominent potential of the approach.

Applying Cell Painting to phytopathogenic filamentous fungi is of paramount importance as we strive to produce the next generation of sustainable products for crop protection. In our study, we focus on the application of these techniques to the Cell Painting images of the fungus *Botrytis cinerea* (Bi et al., 2023). In the experimental setup, 3 fluorescent dyes

are used: Nile Red, Calcofluor (CFW), and Wheat Germ Agglutinin (WGA) to stain lipids, cell wall, and apical cell wall, respectively. Additionally, label-free brightfield microscopy is carried out, resulting in an overall 4-channel image for each condition.

In this paper, a supervised convolutional neural network (CNN)-based model is presented as a proof-of-concept, and the limitations of such an approach are outlined. We further introduce a self-supervised method, which is currently under exploration, emphasizing its advantages in the given problem setting. We also provide an outlook for further studies.

## 2. Image analysis models

In this section, deep learning models for the cellular feature extraction of the filamentous fungi are discussed: a Deep Cosine Metric Learning (DCML) supervised model (Wojke and Bewley, 2018), and a self-supervised DINO-based model (Caron et al., 2021).

### 2.1. Deep Cosine Metric Learning model

The DCML model is a CNN-based feature-extraction model that creates an embedding for each input image, such that samples with similar phenotypes are positioned closer to each other in the feature space, while samples with contrasting characteristics are positioned further apart. To achieve this, the network learns on manually annotated data by optimizing the cosine similarity of the samples with the same label. The learned embeddings can be used for various downstream tasks, such as drug mode of action inference through clustering analysis, or estimation of dose-response profiles (Lejeune et al., 2023). Even though this method proves to be efficient, it has two significant limitations: it requires manually annotated data, which is labor-intensive and prone to human error; and its prediction capabilities hardly go beyond human accuracy.

### 2.2. Self-supervised model

The constraints of the DCML method can be circumvented by using self-supervised models. In this case, the models do not rely on any labels and rather use a surrogate objective to create expressive representations potentially useful for downstream tasks. Since data do not require to be manually labeled, the model is not limited to known classes. Self-supervised methods are currently state-of-the-art, notably in the analysis of natural data. In our study, we focus on adapting the so-called DINO model used for unsupervised feature learning. DINO employs two networks, each receiving the same image as input but with different random transformations. Self-supervision occurs by matching their outputs, which are compared using a cross-entropy loss.

Training a DINO-like model is notoriously data-intensive, and its adaptation to the fungal images is currently ongoing. However, initial preliminary findings, using a pre-trained network for the inference, demonstrate promising outcomes. Figure 1 shows a comparison between compound activities evaluated from the features extracted with DCML and the pre-trained DINO model. In WGA and NileRed image modalities both approaches show similar results, indicating the DINO's robustness. Moreover, by investigating the brightfield image samples at 0.04 (center) and 0.8 (right) drug concentrations, a significant difference in

effects on the fungi can be noted, which is captured by the DINO while the DCML network fails to do so. We expect the performance to improve even further when being specifically trained on fungal data.

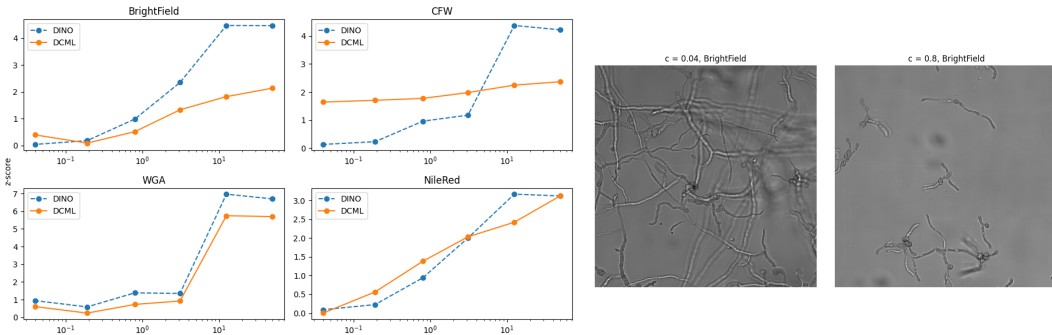

Figure 1: Left: z-scores calculated from the distance to the negative control sample in the feature space obtained with DCML and DINO model plotted against compound concentration. Center, Right: fungi treated with the compound of 0.04 (center) and 0.8 (right) concentrations in the brightfield modality.

## 3. Outlook and conclusion

Self-supervised models have already been successfully tested on Cell Painting data. However, with respect to previous results, our use case shows some distinguishing complex factors. The target organisms are inherently multicellular, developing over time and going through radical changes in their cell biology and global shape, both across different treatments and different species of interest, ranging from isolated round-shaped spores to long branching mycelia. The high sensitivity of organism shape to treatment is one of the most characteristic features of filamentous fungi and has to be carefully taken into account in all morphology-based analyses. Deep-learning models have to be tailored to recognize, characterize, and make abstractions of these differences in shape.

The goal of our analyses is to profile a compound in terms of its mechanism-of-action (MoA), thereby leveraging the bulk of morphological information contained across all fluorescent channels. At the same time, however, the need for a precise characterization of the drug's effects beyond its MoA requires models that are channel-specific and sensitive to the slightest morphological change. Whether this broad spectrum of read-outs can be achieved with a single unifying model, or whether a set of different approaches are needed, is one of the main open questions of our research. Such a finer level of interpretation could potentially be achieved by combining morphology-based features with gene-expression information (under the form of *-omics* data). Cleverly combining these different levels of information on drug effects, possibly by projecting them onto a common latent space (Bao et al., 2022), would allow one to connect two very different levels of cellular responses, greatly enhancing the predictive power of any downstream analyses.

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
