# OpenReview forum: "Cell-painting phenotyping in filamentous fungi: deep learning for image-based drug discovery"
_MIDL.io/2024/Short_Papers — MIDL 2024 Short Papers_

### Official Review · Reviewer_gABy · 2024-04-24

**Confidence:** 3
**Final Rating:** 3.5

**Review:**

Cell painting using multiple dyes is used increasingly commonly for phenotyping. Feature extraction, with deep learning, on these images has become an active area of research including self-supervised learning. This paper compares a supervised metric learning  model with DINO self-supervised learning. The results while being limited show promise of the DINO approach.

---

### Decision · Program_Chairs · 2024-04-26

Accept